# Development of explicit definitions of potentially inappropriate prescriptions for antidiabetic drugs in people with type 2 diabetes: A Delphi survey and consensus meeting

Erwin Gerard[1]*, Derya Bugday[1], Matthieu Calafiore[1,2], Jan Baran[2], Sophie Gautier[3], Heloïse Henry[4], Bertrand Decaudin[4], Madleen Lemaitre[1,5], Nicolas Baclet[1], Paul Quindroit[1], Anne Vambergue[5,6], Jean-Baptiste Beuscart[1]

1 Université de Lille, CHU Lille, ULR 2694 - METRICS: Évaluation des technologies de santé et des pratiques médicales, F-59000 Lille, France, 2 Department of General Practice, University of Lille, Lille, F-59000 Lille, France, 3 Université de Lille, CHU de Lille, UMR-S1172, Center for Pharmacovigilance, Lille, France CHU Lille, Institut de Pharmacie, F-59000 Lille, France, 4 Université de Lille, CHU Lille, ULR 7365 - GRITA - Groupe de Recherche sur les formes Injectables et les Technologies Associées, F-59000 Lille, France, 5 CHU Lille, Department of Diabetology, Endocrinology, Metabolism and Nutrition, Lille University Hospital, F-59000, Lille, France, 6 European Genomic Institute for Diabetes, University School of Medicine, F-59000 Lille, France

* erwin.gerard@univ-lille.fr

## Abstract

### Introduction

Explicit definitions for potentially inappropriate prescriptions (PIPs) are useful for optimizing drug use. The objective of the present study was to validate a list of definitions of PIPs for antidiabetic drugs in a Delphi survey with general practitioners, diabetologists, community pharmacists, hospital pharmacists and pharmacologists from mainland France, Belgium, and Switzerland.

### Methods

The experts gave their opinion on each explicit definition and could suggest new definitions. Definitions with a 1-to-9 Likert score of between 7 and 9 from at least 75% of the participants were validated. The results were discussed during consensus meetings after each round.

### Results

46 participants were recruited, and 38 (82.6%) completed the survey. The Delphi survey resulted in a consensus list of 41 explicit definitions of PIPs for antidiabetic drugs in four groups: (i) the need to temporarily discontinue a medication in the event of acute illness (n = 9; 22%), (ii) the need to review and adjust the dosing regimen (n = 26; 36.6%), (iii) the initiation of an inappropriate drug (n = 3; 7.3%), and (iv) the need for further monitoring of a people with type 2 diabetes (n = 3; 7.3%).

**Data availability statement:** All relevant data are within the manuscript and its Supporting Information files.

**Funding:** This research was funded by PreciDIAB, which is jointly supported by the French National Agency for Research (ANR- 18- IBHU- 0001), by the European Union (FEDER - agreement NP0025517), by the Hauts- de- France Regional Council (agreement 20001891/ NP0025517) and by the European Metropolis of Lille (MEL, agreement 2019_ESR_11). The funders had no role in study design, data collection and analysis, decision to publish, or preparation of the manuscript.

**Competing interests:** The authors have declared that no competing interests exist.

## Conclusions

The list is specific for antidiabetic drugs (other than insulin) for people with type 2 diabetes. This explicit list could be implemented in a clinical decision support system for the automatic detection of PIPs and might help healthcare professionals involved in the management of people living with type 2 diabetes.

## 1 Introduction

The management of antidiabetic drugs (ADs) for people with type 2 diabetes (T2DM) has become more complicated over the last decade. The introduction of new classes of drug (such as sodium/glucose cotransporter 2 inhibitors, dipeptidyl peptidase-4 inhibitors, and glucagon-like peptide-1 receptor agonists) has influenced the choice of first-line treatments and combination therapies. [1–4]. The management of ADs in primary care can be challenging [5–7], and low-quality prescriptions increase the risks of hypoglycaemia and hospital admission [8–11]. National and international guidelines on the appropriate prescribing of ADs are regularly published and updated [1,12]. The application of these guidelines requires an implicit judgement, based on an expert's assessment of (amongst others) research data, the clinical circumstances, and the patient's preferences. An implicit approach is effective in reduce inappropriate prescriptions but has a number of practical limitations: its application is time-consuming and requires specialized knowledge and skills [13]. The multidisciplinary management of people living with T2DM has been recommended by many experts [14–16]. Several studies have shown that non-diabetic physicians and pharmacists face barriers to implementing national and international guidelines and optimizing prescriptions for people living with T2DM. These barriers including limited resources, time and workload constraints, and a lack of skills and knowledge of diabetes management [17–19].

An explicit approach is a complementary way of reducing inappropriate prescriptions [13]. The International Group for Reducing Inappropriate Medication Use & Polypharmacy recommended the combination of explicit and implicit approaches [20]. Explicit tools are based on predefined rules for analyzing drug prescriptions, do not require the intervention of an expert [21], and are therefore easier to implement in general and to automate in computerized decision support systems (CDSSs) [22,23]. However, neither implicit nor explicit approaches should be considered as a universal solution to inappropriate prescribing. Implicit and explicit approaches should be combined, which underlines the importance of treatment personalization according to individual patient characteristics [1,13].

Most tools for assessing potentially inappropriate prescriptions (PIPs) were developed for use with older people; lists of PIPs (such as Beer's criteria and the STOPP/ START tool) are widely used on a routine basis [24,25]. The effectiveness of explicit tools (such as lists of PIPs) has been well documented [26,27]. However, these tools are not specific for ADs management in people living with T2DM. PIP-AD can lead to serious clinical consequences such as hypoglycemia, ketoacidosis, or pancreatitis,

underlining the importance of developing AD-specific criteria ketoacidosis [28–30]. Our recent systematic review highlighted a lack of consensus on definitions of PIPs of ADs (hereafter referred to as PIP-ADs): the definitions were heterogenous and focused primarily on at-risk situations related to (i) biguanide prescriptions in people with renal dysfunction, and (ii) sulfonylurea prescriptions in older adults [31]. A qualitative study of 30 healthcare professionals (general practitioners, diabetologists, and pharmacists) listed 38 new explicit definitions of PIP-ADs [32]. This qualitative study gathered the opinions of healthcare professionals but did not establish a consensus on the use of these rules in clinical practice [33,34]. In this study, we focused on non-insulin AD. Insulin was excluded because its prescribing is highly individualized, requiring titration and frequent patient-specific adjustments, which makes it less suitable for standardized explicit definitions [1,35].

The objective of the present study was to validate this list of definitions by using a Delphi consensus method, with participants from mainland France, Belgium, and Switzerland.

## 2 Material and methods

### Study design

A Delphi survey is a structured group interaction that relies on questionnaires (rather than face-to-face meetings) to facilitate communication among participants [34,36]. This iterative process gathers the opinions of a panel of experts on a predefined list of items. Each iteration is referred to as a "round," during which participants complete a questionnaire anonymously. Delphi surveys are frequently recommended for the development of healthcare guidelines. [37,38]. In the present study, we conducted a Delphi survey on a list of explicit definitions of PIP-ADs, as described in our study protocol [39]. This method has already been used in the context of potentially inappropriate prescribing, particularly in the fields of geriatrics and infectious diseases [25,40]. The explicit definitions were identified through a systematic review of the literature and qualitative study, as described in the study protocol [39]. The study included participants from France, Belgium and Switzerland. The present Delphi survey was reported in accordance with the Accurate Consensus Reporting Document (ACCORD) guidelines [41] (S1 Table).

### Steering committee

The study's methodological principles and results at each step were validated by a steering committee. The steering committee comprised a diabetologist (AV), a general practitioner (MC), a pharmacist (BD), and a pharmacologist (SG).

### Ethical approval

This study was approved by an independent ethics committee at the University of Lille (Lille France; reference: 2024−073). The healthcare professionals agreed to participate in the study and consented to the collection of their personal data, in accordance with the European Union's General Data Protection Regulation. All participants provided electronic written informed consent before participating in the Delphi survey. They were informed that their answers would remain anonymous and had to actively confirm their agreement by checking a consent box prior to accessing the questionnaire.

### Recruitment of participants

We sought to recruit 40–50 participants from France, Belgium, and Switzerland between February 1st, 2024 and April 1st, 2024. We expected to achieve a response rate of at least 80% from the recruited participants. Five groups of participants were identified, with the following target distribution: general practitioners (25%), diabetologists (25%), community pharmacists (16.6%), hospital pharmacists (16.6%), and pharmacologists (16.6%). Participants were identified through the professional networks of the steering committee. Each steering committee member reached healthcare professional who were specifically invested in AD prescribing and then contacted them by email.

**The online platform**

The online survey was prepared on the SmartSurvey™ platform (https://www.smartsurvey.com). An introduction page presented the study's objectives and procedures via a summary and an instructional video. Information about the European Union's General Data Protection Regulation was made available. The SmartSurvey™ platform generated e-mails and reminders for the Delphi panel members. Each participant had a personal link for accessing the online survey.

**Data collection**

Before they accessed the Delphi survey questionnaire, the participants completed a questionnaire and indicated their age, sex, year of graduation, specialty, and place of practice. To avoid fatigue bias, the explicit definitions were always presented in random order. Participants did not receive individual feedback reports between rounds. For items carried forward to round 2, we presented aggregate feedback together with a synthesis of arguments during the first consensus meeting to support reflective re-assessment. To minimise respondent burden, these materials were not embedded in the round-2 online questionnaire; non-attendees received the same quantitative summaries and a written synopsis of the discussion.

**The Delphi process**

**2.1.1 Rounds 1 and 2.** The Delphi process consisted of two questionnaires rounds. In round 1, the participants rated the explicit definitions on a Likert scale ranging from 1 to 9, where 1 corresponded to "I totally disagree" and 9 corresponded to "I totally agree". The participants were allowed to choose a "no opinion" answer. For each explicit definition, the participants had the opportunity to add a free-text comment. Lastly, after responding to all the explicit definitions, the participants could suggest new ones.

In round 2, the participants again rated the explicit definitions on the same Likert scale ranging from 1 to 9, plus a "no opinion" option. The explicit definitions voted on in the second round included (i) definitions that had not been validated in round 1; (ii) new suggested definitions considered to be relevant by the consensus meeting; and (iii) reformulated definitions. For the new definitions, participants had the opportunity to add a free-text comment. Participants who did not respond in round 1 were not invited to round 2.

**2.1.2 Consensus meetings.** One-hour videoconference consensus meetings were organized by four investigators (E.G., D.B., P.Q., and J.-B.B.) at the end of each round. The study's steering committee and available participants attended the consensus meetings. The researchers presented the results, including a summary of all the comments and discussions about the explicit definitions, and discussed them with the participants.

The objectives of the consensus meeting at the end of round 1 were to present the validated and rejected definitions, consider possible reformulations, and discuss the addition of new definitions. The objectives of the consensus meeting at the end of round 2 were to present the validated and rejected definitions, discuss possible reformulations, and consider possible reformulations, and discuss definitions that had not been validated or rejected. The Wooclap application (https://wooclap.com) was used to allow participants to share their responses and views and vote to the explicit definitions suggested during the consensus meeting. The votes in the consensus meetings were on a three-point scale: "agree," "disagree," or "no opinion." The members of the steering committee were able to share their view s during the consensus meetings but did not vote. At the end of each consensus meeting, a report was sent to all the participants.

**Definition of the consensus**

During the survey, a definition was considered to have been validated if (i) at least 75% of the participants had given it a Likert scale score of between 7 and 9 (inclusive), and (ii) less than 15% of the participants had given it a Likert scale score of between 1 and 3 (inclusive). A definition was considered to have been rejected if at least 75% of the participants had

given it a Likert scale score of between 1 and 3, and less than 15% of the participants had given it a Likert scale score of between 7 and 9. In other cases, the explicit definitions were considered to be indeterminate as they were neither validated nor rejected by the panel [38,42]. If necessary, votes conducted during consensus meetings will use a three-point scale: "Agree", "Disagree", or "No opinion", the consensus threshold criterion was 75% [38,42].

## Data analysis

Statistical analyses were performed using R software (version 4.1.2) [43]. Fig 1 was generated in R (*sf*, *tmap*) using Natural Earth administrative boundaries (public domain). No proprietary basemaps were used. Qualitative variables were expressed as the frequency (percentage) for each category. Quantitative variables were expressed as the median (range). Definitions were classified as validated, rejected, or indeterminate. Each free-text comment and the new suggestions were analysed independently by two researchers (E.G. and D.B.). Disagreements were resolved by a third researcher (J.-B.B.),

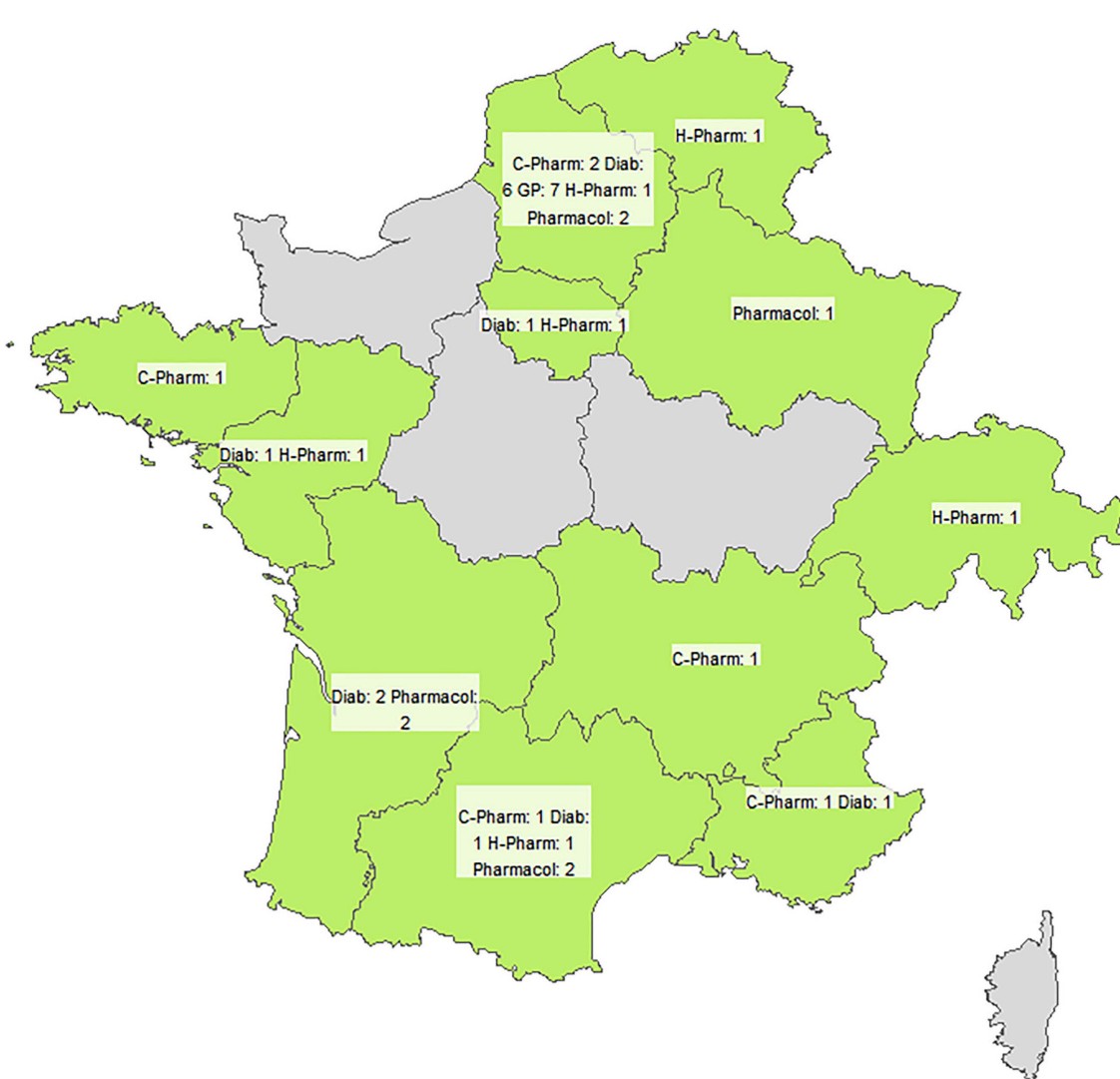

**Fig 1. Geographic distribution of survey respondents.**

if necessary. All participants' responses were weighted equally, irrespective of professional expertise, in line with standard Delphi methodology [41].

## 3 Results

### Participants

For this Delphi survey, 46 participants were recruited, and 38 (82.6%) completed the full survey: 7 (18.4%) general practitioners, 12 (31.6%) diabetologists, 6 (15.8%) hospital pharmacists, 6 (15.8%) community pharmacists, and 7 (18.4%) pharmacologists (Table 1). With regard to the geographical distribution, there were participants from 8 of the 13 regions in mainland France, 1 participant from Belgium, and 1 participant from Switzerland (Fig 1).

### Delphi survey

Fig 2 illustrates the flow of participants across the two Delphi rounds and the evolution of the explicit definitions over time. The first Delphi round took place between April 8th and May 19th, 2024. The data collected was analysed between May 19th, 2024 and May 27th, 2024. Of the 46 participants recruited, 41 (89.1%) completed the first round. In round 1, 26 of the 38 explicit definitions were validated, 12 were considered to be indeterminate, and none were rejected. The participants submitted a total of 231 comments and 4 new explicit definitions. At the consensus meeting at the end of round one (on May 27th, 2024), 13 definitions were reformulated, and 3 new definitions were included in the second round.

The second round took place between May 31st and June 23rd, 2024. The data collected was analysed between June 23rd, 2024 and July 2nd, 2024. Of the 46 participants recruited, 38 (82.6%) completed the survey. In round 2, 13 definitions were validated, 3 were considered to be indeterminate, and none were rejected. The participants submitted a total of 6 comments about the new definitions. During the consensus meeting at the end of round 2 (on July 2nd, 2024), the group decided to include two indeterminate definitions and exclude one definition. Ultimately, two definitions were reformulated.

### The consensus list of explicit definitions of PIP-ADs

The Delphi consensus list contained 41 explicit definitions of PIP-ADs, in four groups (Table 2). The first group contained definitions (n = 9; 22%) used to assess the need to temporarily discontinue an AD, e.g., "It is necessary to assess the need to temporarily discontinue metformin for 48 hours if the patient is scheduled to receive an iodinated contrast agent". The second group (n = 26; 36.6%) contained definitions used to assess the need to optimize and adjust treatment dosing under specific circumstances, e.g., "It is necessary to review and optimize the prescription of metformin, a GLP-1 receptor agonist, or a DPP4 inhibitor in a person whose creatinine level has increased by a factor of 1.5 in 7 days". The third group (n = 3; 7.3%) contained definitions used to define inappropriate initiation, e.g., "Do not initiate treatment with a hypoglycaemic drug (repaglinide or a hypoglycaemic sulphonamide) in a person aged 75 or over". The fourth group (n = 3; 7.3%) contained definitions used to assess the need for further monitoring of a person living with T2DM, e.g., "The HbA1c level should be monitored every 3 months in a person receiving an antidiabetic drug (other than insulin)".

Table 1. Characteristics of the survey participants.

| | Participants n = 38 (100%) | General practitioners n = 7 (18.4%) | Diabetologists n = 12 (31.6%) | Hospital pharmacists n = 6 (15.8%) | Community pharmacists n = 6 (15.8%) | Pharmacologists n = 7 (18.4%) |
|---|---|---|---|---|---|---|
| Median age, years [range] | 44 [26; 68] | 46 [32; 68] | 41 [31; 68] | 37.5 [29; 48] | 43.5 [26; 57] | 49 [33; 60] |
| Median years of qualification [range] | 2008 [1982; 2022] | 2008 [1982;2021] | 2011 [1983;2021] | 2017 [2003; 2022] | 2006 [1991; 2021] | 2004 [1993; 2019] |
| Females, n (%) | 23 (60.5%) | 1 (14.0%) | 8 (67.0%) | 5 (83%) | 3 (50.0%) | 6 (86.0%) |

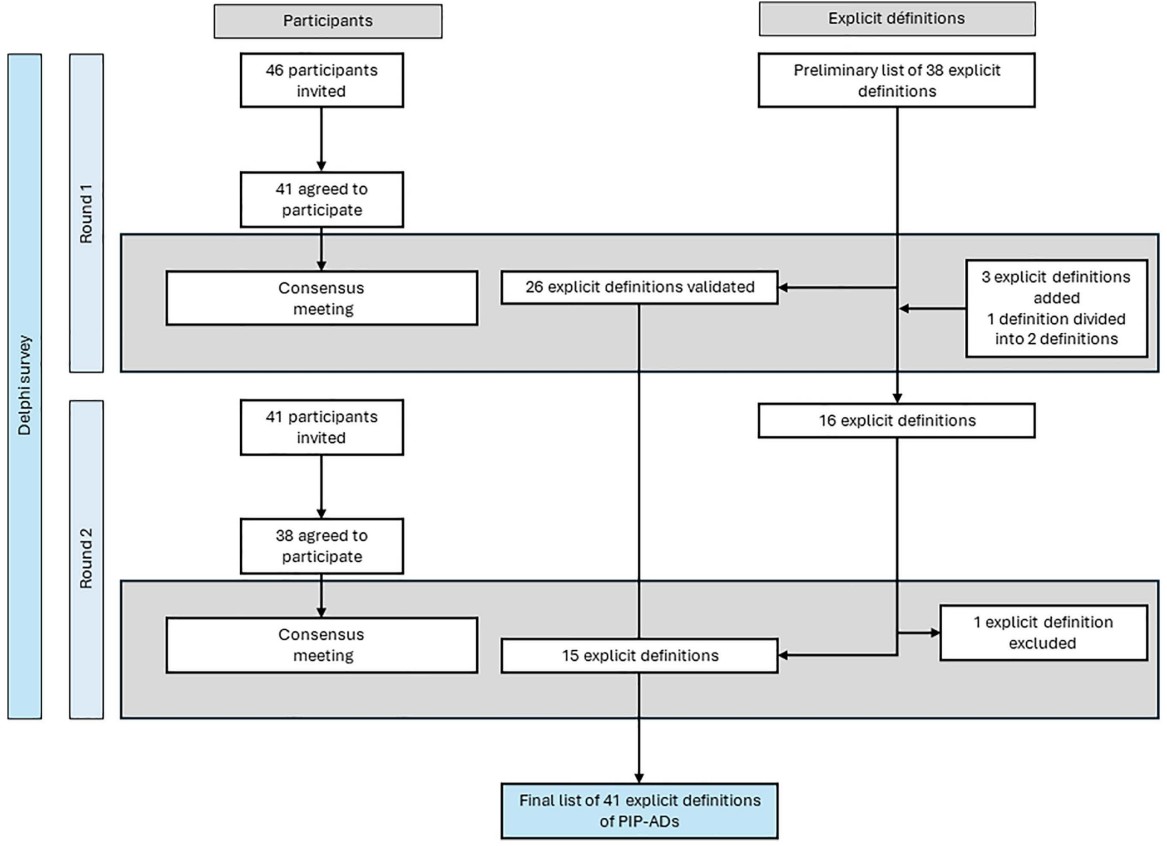

**Fig 2. Diagram showing the flow of participants and explicit definitions of potentially inappropriate prescriptions of antidiabetic drugs during the Delphi survey and the consensus meetings.**

## 4 Discussion

### Main findings

As mentioned above, the objective of the present Delphi study was to validate a new list of explicit definitions of PIP-ADs. A total of 46 experts healthcare professionals from clinical domains were recruited to validate a list of 41 explicit definitions. The participation rate was 82.6%. Explicit lists of PIPs facilitate the detection of at-risk prescriptions. Some definitions were highly specific (e.g., the combination of metformin and dehydration). These originated from the proposals collected during the preliminary qualitative study [32] and from expert contributions during the Delphi rounds, and were subsequently discussed during the consensus meetings to ensure alignment with national and international guidelines. While certain PIP scenarios may be uncommon, others (e.g., sulfonylurea use in older adults) are frequent in routine practice, reinforcing the importance of clear explicit definitions. For non-specialists, this new list might raise awareness of situations that might require the optimization of prescriptions among people living with T2DM.

### Perspectives for the application of explicit definitions of PIP-ADs

Explicit tools for use with populations of older adults have been studied extensively, such as the Beers criteria, the STOPP/START criteria, the EU(7)-PIM list, and the PRISCUS list [24,25,44–47]. The use of such lists is known to improve the appropriateness of prescriptions [26], decrease the risk of adverse drug reactions [48], reduce healthcare costs [49],

**Table 2. Explicit definitions of potentially inappropriate prescriptions for antidiabetic drugs in people with type 2 diabetes.**

**It is necessary to assess the need to temporarily discontinue…**

1. It is necessary to assess the need to temporarily discontinue the prescription of an antidiabetic drug (other than insulin) in a patient with a diagnosis of acute pancreatitis, including suspected diagnosis of acute pancreatitis (elevated lipase levels, abdominal pain, etc.).

2. It is necessary to assess the need to temporarily discontinue metformin for 48 hours when the patient is scheduled to receive an iodinated contrast agent.

3. It is necessary to assess the need to temporarily discontinue an antidiabetic drug (other than insulin) in a patient undergoing surgery under general anaesthesia, the day before the operation, on the day of the operation, and the day after operation.

4. It is necessary to assess the need to temporarily discontinue an antidiabetic drug (other than insulin) in the context of acute digestive disorders (nausea, diarrhoea, and vomiting).

5. It is necessary to assess the need to temporarily discontinue an antidiabetic drug (other than insulin) in a patient with acute liver failure.

6. It is necessary to assess the need to temporarily discontinue an antidiabetic drug (other than insulin) in a patient exhibiting signs of infection (fever, sepsis, the concurrent use of antibiotic, etc.).

7. It is necessary to assess the need to temporarily discontinue an SGLT-2 inhibitor in a patient with a bacterial skin infection localized in the pelvic or genital area.

8. It is necessary to assess the need to temporarily discontinue a hypoglycaemic drug (repaglinide or a sulfonylurea) in a patient with a capillary blood glucose level below 0.8 g/L at the time of administration.

9. It is necessary to assess the need to temporarily discontinue an SGLT-2 inhibitor in a patient with severe (grade 3 or grade 4), unstable, peripheral arterial disease of the lower limbs.

**It is necessary not to initiate…**

10. It is necessary not to initiate treatment with an antidiabetic drug in a patient with a non-confirmed diagnosis of type 2 diabetes (two venous blood glucose values of 1.26 g/L or 1 venous blood glucose value of 2 g/L).

11. It is necessary not to initiate treatment with a hypoglycaemic drug (repaglinide or sulfonylurea) in a patient over 75 years old.

12. It is necessary not to initiate treatment with a hypoglycaemic drug (repaglinide or sulfonylurea) in a patient with an HbA1c level below 7%.

**It is necessary to review and optimize…**

13. It is necessary to review and optimize the prescription of an antidiabetic drug in a patient who does not tolerate the prescribed treatment.

14. It is necessary to review and optimize the prescription of an antidiabetic drug (other than insulin) in a dependent patient (with motor disorders, cognitive impairment, ALD score, AGGIR, etc.) who does not receive home care or who is not institutionalized.

15. It is necessary to review and optimize the strategy for two- or three-drug combination treatments that include a hypoglycaemic drug (repaglinide or sulfonylurea) in a patient with an HbA1c level below 7%.

16. It is necessary to review and optimize the prescription of an antidiabetic drug (other than insulin) in a patient with a history of diabetic ketoacidosis.

17. It is necessary to review and optimize the prescription of a DPP-4 inhibitor or GLP-1 analogue in a patient with a pancreatic disorder.

18. It is necessary to review and optimize the prescription of an antidiabetic drug (other than insulin) if it is clearly stated in the patient's medical record that they refuse to take this drug.

19. It is necessary to review and optimize the prescription of an antidiabetic drug (other than insulin) in a patient taking more than 8 prescription drugs and with an HbA1c level below 7%.

20. It is necessary to review and optimize the prescription of a combination of two antidiabetic drugs (other than insulin) from the same therapeutic class.

21. It is necessary to review and optimize the prescription of an injectable antidiabetic drug if it is difficult to administer an injectable drug to a patient (fear of needles, etc.).

22. It is necessary to review and optimize the prescription of an oral antidiabetic drug in a patient who has difficulty swallowing tablets.

23. It is necessary to review and optimize the prescription of metformin, a GLP-1 analogue, or a DPP-4 inhibitor in a patient with a creatinine level that has increased 1.5-fold in 7 days.

24. It is necessary to review and optimize the prescription of a GLP-1 analogue + DPP-4 inhibitor combination.

25. It is necessary to review and optimize the prescription of an SGLT-2 inhibitor in a patient with a genitourinary condition (a lower urinary tract infection in men or recurrent (≥ 4 times per year) lower urinary tract infections in women, an upper urinary tract infection, urinary tract disorders, genital fungal infections, etc.) or a history of genitourinary problems.

26. It is necessary to review and optimize the prescription of a weight-loss promoting drug (SGLT-2 inhibitor or GLP-1 analogue) in a patient with a BMI < 18 kg/m².

27. It is necessary to review and optimize the prescription of metformin in a patient with a condition that may lead to hypoxia (respiratory disease, sleep apnoea syndrome, a patient on oxygen therapy, etc.).

28. It is necessary to review and optimize the prescription of a repaglinide + sulfonylurea combination.

*(Continued)*

**Table 2.** (Continued)

| It is necessary to review and optimize… |
|---|
| 29. It is necessary to review and optimize the prescription of a hypoglycaemic drug (repaglinide or sulfonylurea) in a patient on a low-calorie diet (for weight loss/slimming). |
| 30. It is necessary to review and optimize the prescription of a hypoglycaemic drug (repaglinide or sulfonylurea) in a patient reporting symptomatic hypoglycaemia (dizziness, sweating, palpitations, intense hunger, weakness, etc.). |
| 31. It is necessary to review and optimize the prescription of an antidiabetic drug (other than insulin) in a patient with chronic liver disease. |
| 32. It is necessary to review and optimize the prescription of a GLP-1 analogue in a patient with a history of gallstones. |
| 33. It is necessary to review and optimize the prescription of a hypoglycaemic drug (repaglinide or sulfonylurea) in a patient with a BMI > 30 kg/m². |
| 34. It is necessary to review and optimize the prescription of an antidiabetic drug (other than insulin) from the DPP-4 inhibitor and GLP-1 analogue classes in a patient with a pancreatic condition. |
| 35. It is necessary to review and optimize the prescription of metformin in a patient who might become dehydrated and will not be able to received medical care within 48 hours (e.g., water fasting, travel to hot countries, multiday hikes, long boat trips during summer, etc.). |
| 36. It is necessary to review and optimize the prescription of an antidiabetic drug (other than insulin) in a pregnant woman living with type 2 diabetes. |
| 37. It is necessary to review and optimize the prescription of a GLP-1 analogue in a patient with a confirmed diagnosis of malnutrition. |
| 38. It is necessary to review and optimize the prescription of an antidiabetic drug (other than insulin) from the SGLT-2 inhibitor class in a patient with pancreatic insufficiency (a risk of diabetic ketoacidosis). |
| **It is necessary to monitor…** |
| 39. It is necessary to monitor the estimated glomerular filtration rate (eGFR) of a patient on an antidiabetic drug (other than insulin) at least once a year. |
| 40. It is necessary to monitor the HbA1c level of a patient on antidiabetic drug (other than insulin) every 3 months. |
| 41. It is necessary to monitor the patient's adherence to their treatment and their involvement in a patient education program. |

and increase health-related quality of life [50]. These benefits have been observed in primary care [51–53], in nursing homes [54,55] and in hospital settings [26,56,57]. These studies also highlighted the importance of collaboration between the various healthcare professionals involved in patient care, i.e., general practitioners, pharmacists, and nurses.

Building on this background, our research study was prompted by the need to develop explicit tools specific to the management of ADs. The explicit list obtained in the present study is specific for the management of ADs (excluding insulin) for people living with T2DM. Polypharmacy is prevalent among older adults with diabetes, due to the many comorbidities, and the management of AD prescriptions in primary care can be challenging [5]. Faquetti et al.'s study of a primary care cohort in the UK showed that 39.6% of people with T2DM aged 65 or over had one or more PIPs of non-insulin ADs [58]. It has been shown that PIP-ADs are associated with a greater risk of adverse drug reactions, such as hypoglycaemia, pancreatitis, and ketoacidosis [28–30]. This new list of PIP-ADs might notably be of help to healthcare professionals who manage people living with T2DM, especially professionals who are not diabetes specialists or when patients are managed in an isolated, non-multidisciplinary context.

This experience with explicit lists of PIPs among older adults therefore raises the question of how our PIP-AD list should be implemented. Firstly, the relevance of our list should be evaluated in real life (as performed widely among older adults), in order to determine the frequency of PIP-ADs and their association with adverse drug reactions [59]. Our list could then be implemented in a CDSS for the automatic detection of PIP-ADs in electronic medical records in primary care [60] or in a hospital setting [23]. Other explicit lists (such as the STOPP-START criteria and the REMEDI[e]S list) have been translated into seminatural language for use in CDSSs [61,62]. Explicit definitions are machine-readable, but CDSS automation depends on local data and coding. Rules based on structured EHR fields (e.g., age, comorbidities, laboratory results) can be implemented directly, whereas performance varies with how drugs and conditions are encoded (active-ingredient vs ATC ATC 5th-level mapping, specific ICD-10 or SNOMED codes). By contrast, concepts such as dehydration or recurrent hypoglycaemia may exist only in free text, which limits automation [63].These tools have also demonstrated their value for detecting situations with a risk of adverse drug reactions and could therefore be used to

automatically detect PIPs by various healthcare professionals: general practitioners [64,65], community pharmacists [66], hospital pharmacists [67], and nurses [68]. However, a recent systematic review showed that only few studies have investigated the detection of PIP-ADs with a CDSS and that the results varied according to the operating context [69]. It is therefore necessary to assess the operating contexts in which these rules will be implemented or in which these tools will be used [70]. In the future, we shall need to assess the definitions' relevance with regard to their context of use and thus the definition of the right information for the right person in the right intervention format and the right channel, and at the right time (to avoid alert fatigue) [71–73]. Lastly, it would be interesting to integrate implicit definitions (i.e., from national or international guidelines) with explicit definitions; as already shown for the REMEDI[e]S list, an implicit-explicit approach might prevent and counter inappropriate prescribing [20,44].

## Strengths and limitations

The present work was a continuation of previous research [31,39] and was based on a method that has frequently been used to generate and validate new knowledge through a consensus process [40,74]. The Delphi survey was carried out in line with the ACCORD guidelines [41]. This survey was conducted using a multidisciplinary expert panel, complied with the guidelines on the management of people living with T2DM, and involved participants from several disciplines: general practitioners, diabetologists, community pharmacists, hospital pharmacists, and pharmacologists [14]. The participants came from different regions of mainland France, Switzerland, and Belgium. Although the final distribution of participants differed slightly from the initial target (e.g., 18.4% GPs instead of 25%), the panel remained multidisciplinary, ensuring that the consensus reflected the perspectives of all major professional groups. The proportion of female GPs (14%) reflects our recruited sample and may not be fully representative of the GP population in France. Nevertheless, this discrepancy is unlikely to have influenced the validity of our findings, as the Delphi methodology aims to synthesize expert opinions rather than to achieve population-level representativeness.

The study had some limitations. Given that the study was conducted in France and even though two of the participants came from abroad (Belgium and Switzerland), it is possible that certain pharmaceutical specialties subject to PIPs were not taken into account because they are not available in France (e.g., thiazolidinedione drugs) [75,76]. This work is not intended to limit the use of these lists to a single region of the world; our definitions can be adapted to international contexts by considering differences in drug availability and prescribing practices. For example, the Beers criteria were initially developed in the USA and the STOPP/START criteria were initially developed in Ireland; the two lists are now applied around the world [77–80]. Another limitation is that the list does not cover all antidiabetic drug classes (e.g., thiazolidinediones) and excludes insulin. Future research should extend these explicit definitions to a broader range of drugs, including insulin.

## 5  Conclusion

Using a structured Delphi method and a consensus meeting with a large number of healthcare professionals from France, Switzerland and Belgium, we developed the first consensus list of explicit definitions of PIP-ADs for people living with T2DM. The participation rate was high. This tool may help healthcare professionals involved in the management of people living with T2DM, and particularly those who are not diabetes specialists. The list could be implemented in a CDSS for the automatic detection of PIP-ADs in electronic medical records. The next step will be to validate these explicit definitions in real-world clinical settings to assess their feasibility, reliability, and impact on prescribing practices, particularly in the context of clinical decision support systems.

## Supporting information

**S1 Table.  The ACCORD checklist for the reportingof consensus methods.**
(XLSX)

## Acknowledgments

The authors thank all the Delphi survey participants: Anne Claire Leguillou, Hauts-de-France, France; Caroline Sanz, Occitanie, France; Celine Michel, Belgique; Charles Cauet, Nord-Pas-de-Calais, France; Christel Roland, Occitanie, France; Christine Lemaire, Nord-Pas-de-Calais, France; Christophe Berkout, Hauts-de-France, France; Dany Delberghe, Hauts-de-France, France; David Wyts, Hauts-de-France, France; Didier Gouet, Nouvelle-Aquitaine, France. Emeline Ranson, Hauts-de-France, France; Emmanuelle Lecornet Sokol, Île-de-France, France. Felicia Ferrera, Provence-Alpes-Côte d'Azur, France; Florence Baudoux, Hauts-de-France, France; Florent Verdier, Nouvelle-Aquitaine, France; François Loez, Hauts-de-France, France; Gabrielle Lisembard, Hauts-de-France, France; Guillaume Gory, Occitanie, France; Haleh Bagheri, Occitanie, France; Heloise Henry, Hauts-de-France, France; Isabelle Geiler et Claire Boulanger, Hauts-de-France, France; Jean Didier Bardet, Auvergne-Rhône-Alpes, France. Jean Luc Faillie, Occitanie, France; Johanna Béné, Hauts-de-France, France; Léa Sohl Dost, Suisse; Linda Humbert, Hauts-de-France, France; Ludovic Willems, Hauts-de-France, France; Marie Laure Laroche, Nouvelle-Aquitaine, France; Marion Bonsergent, Pays de la Loire, France; Matthieu Chabot, Hauts-de-France, France; Maxence Decroocq, Hauts-de-France, France; Muriel Grau, Nouvelle-Aquitaine, France; Nadine Petitpain, Grand Est, France; Ninon Foussard, Île-de-France, France; Patrice Darmon, Provence-Alpes-Côte d'Azur, France; Patrick Hindlet, Île-de-France, France; Remi Hennevin, Hauts-de-France, France; Sarah Tournayre, Occitanie, France; Simon Rivart, Hauts-de-France, France; Valérie Gras, Hauts-de-France, France; Jessy Trupin, Hauts-de-France, France. The authors thank David Fraser (Biotech Communication SARL, Ploudalmézeau, France) for copy-editing assistance.

## Author contributions

**Conceptualization:** Erwin Gerard, Paul Quindroit, Jean-Baptiste Beuscart.

**Methodology:** Erwin Gerard, Nicolas Baclet, Paul Quindroit, Jean-Baptiste Beuscart.

**Project administration:** Jean-Baptiste Beuscart.

**Supervision:** Matthieu Calafiore, Jan Baran, Sophie Gautier, Bertrand Decaudin, Anne Vambergue, Jean-Baptiste Beuscart.

**Validation:** Matthieu Calafiore, Jan Baran, Sophie Gautier, Heloïse Henry, Bertrand Decaudin, Madleen Lemaitre, Anne Vambergue, Jean-Baptiste Beuscart.

**Writing – original draft:** Erwin Gerard, Derya Bugday, Paul Quindroit, Jean-Baptiste Beuscart.

**Writing – review & editing:** Erwin Gerard, Derya Bugday, Matthieu Calafiore, Jan Baran, Sophie Gautier, Heloïse Henry, Bertrand Decaudin, Madleen Lemaitre, Nicolas Baclet, Paul Quindroit, Anne Vambergue, Jean-Baptiste Beuscart.

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
