## [Decision Letter · Decision Letter 0]

24 Jul 2025

PONE-D-25-27267Development of explicit definitions of potentially inappropriate prescriptions for antidiabetic drugs in people with type 2 diabetes: a Delphi survey and consensus meeting.PLOS ONE

Dear Dr. Gerard,

Thank you for submitting your manuscript to PLOS ONE. After careful consideration, we feel that it has merit but does not fully meet PLOS ONE’s publication criteria as it currently stands. Therefore, we invite you to submit a revised version of the manuscript that addresses the points raised during the review process.

We look forward to receiving your revised manuscript.

Kind regards,

Sairah Hafeez Kamran, PhD

Academic Editor

PLOS ONE

Journal Requirements:

2. Please ensure that you have specified a) Did participants provide their written or verbal informed consent to participate in this study?

- In consent please state in Ethics Method section and manuscript if it is written or verbal. If consent was verbal, please explain a) why written consent was not obtained, b) how you documented participant consent, and c) whether the ethics committees/IRB approved this consent procedure.

 “This research was funded by PreciDIAB, which is jointly supported by the French National Agency for Research (ANR- 18- IBHU- 0001), by the European Union (FEDER - agreement NP0025517), by the Hauts- de- France Regional Council (agreement 20001891/NP0025517) and by the European Metropolis of Lille (MEL, agreement 2019_ESR_11).”

5. We note that Figure 1 in your submission contain map/satellite images which may be copyrighted. All PLOS content is published under the Creative Commons Attribution License (CC BY 4.0), which means that the manuscript, images, and Supporting Information files will be freely available online, and any third party is permitted to access, download, copy, distribute, and use these materials in any way, even commercially, with proper attribution. For these reasons, we cannot publish previously copyrighted maps or satellite images created using proprietary data, such as Google software (Google Maps, Street View, and Earth). For more information, see our copyright guidelines: http://journals.plos.org/plosone/s/licenses-and-copyright.

Additional Editor Comments:

Kindly rectify all grammatical errors.

Reviewers' comments:

Reviewer's Responses to Questions

**Comments to the Author**

1. Is the manuscript technically sound, and do the data support the conclusions?

Reviewer #1: Yes

Reviewer #2: Partly

Reviewer #3: Yes

2. Has the statistical analysis been performed appropriately and rigorously? 

Reviewer #1: N/A

Reviewer #2: N/A

Reviewer #3: Yes

3. Have the authors made all data underlying the findings in their manuscript fully available?

Reviewer #1: Yes

Reviewer #2: Yes

Reviewer #3: Yes

4. Is the manuscript presented in an intelligible fashion and written in standard English?

Reviewer #1: Yes

Reviewer #2: No

Reviewer #3: Yes

5. Review Comments to the Author

Reviewer #1: The paper was generally well written.

Background: Captures the essentials of what the survey is about and the general aim of the paper.

Methodology:Sound

Results and Discussion: Presented in a logical and simplified manner.

Conclusion:It is a reflection of the aims and objectives that the authors sought to obtain.

Authors should declare if they received funding or not for this work.

Reviewer #2: Thank you for submitting this article in an important area. Overall, your work was well reported and provided an interesting read. However, I have a few comments/ suggestions that can improve clarity.

Introduction. You have set the scene well, but I think that there is the opportunity to acknowledge that neither an implicit nor explicit approach to reducing potentially inappropriate medicines is a "fix all". I would also liked to have seen some reference to personalisation of treatment (inferred but not explicit in your description of implicit interpretation on guidelines).

Line 57- "physicians and pharmacists face barriers to implementing guidelines..." - be explicit that you are referring to national and international guidance on prescribing of ADs.

Method

Lines 86-87 Please also include that the explicit definitions were identified by literature as described in the study protocol.

Lines 106-107 "Participants were recruited via email". How were these participants identified in the first instance? Were they from existing relationships within the steering group or did you reach out to special interest groups, etc.

Line 117-"the participants had to complete a questionnaire and indicate" refine to "participants completed a questionnaire and indicated.."

lines 122-132 Were participants provided with an individual report and the start of round 2 to inform them how their responses compared to others participating in the survey? Did you apply analyses e.g. a mixed effects regression model to the data. If not, why not?

Line 155-6 "In other cases, the explicit definitions were considered to be indeterminate (neither validated

156 nor rejected have been neither e in an state"... this sentence does not make sense.

Table 1

Median years since qualification [range] 2008 [1982; 2022]. The median years from qualification is not reported; this is the median YEAR OF qualification.

Figure 1. Whilst this is interesting to see which areas of France were represented in the Delphi study, it would also be interesting to see which healthcare professionals from these regions also participated. you report that you have one Swiss and one Belgian participant, but it would have been interesting to see what type of healthcare professional this was, e.g. GP, community pharmacist, etc.

Line 175 I would open this paragraph describing figure 2 and then outline the timeframe. Currently the reference style made me think the figure also contained dates and not the flow of participants and the evolution of the definitions.

Line 183 "2024.Of " Space required between the end of one sentence and start of another

Discussion

Line 217 "Polypharmacy is very prevalent"- do not need "very"

Line 225 "Explicit tools for use with populations of older adults have been studied extensively"- give examples

Lines 232-250 It feels more organic to lead with existing PIP tools then your tool, highlighting the differences.

Lines 259-262- how can this limitation be adapted for an international audience?

Line 273 Acknowledgments. These seems rather extensive. Other Delhi publications have thanked the participants collectively.

Thank you again for submitting your work for publication.

Reviewer #3: Manuscript Title: Development of explicit definitions of potentially inappropriate prescriptions for antidiabetic drugs in people with type 2 diabetes: a Delphi survey and consensus meeting.

General Assessment:

This study presents a well-structured Delphi consensus approach to developing explicit definitions for potentially inappropriate prescriptions (PIPs) of antidiabetic drugs (ADs) in type 2 diabetes mellitus (T2DM). The topic is clinically relevant, given the increasing complexity of T2DM management and the risks associated with inappropriate prescribing. The methodology is robust, and the results are clearly presented. However, some aspects require clarification or improvement before publication. My comments are enumerated below.

Point-by-Point Comments

1. Title and Abstract

• Title: Appropriate and clear.

• Abstract: The abstract succinctly summarizes the study’s objectives, methods, and key findings.

2. Introduction

• Strengths:

o Clearly outlines the need for explicit PIP definitions in T2DM management.

o Justifies the study by highlighting gaps in existing tools (e.g., lack of AD-specific criteria).

• Suggestions:

o Briefly mention the clinical consequences of PIP-ADs (e.g., hypoglycemia, ketoacidosis) earlier to strengthen rationale.

o Clarify why insulin was excluded from the scope (is it due to different prescribing dynamics?).

3. Methods

• Overall: Well-described and methodologically sound.

• Specific Comments:

o Participant Recruitment:

The target distribution (25% GPs, 25% diabetologists, etc.) is appropriate, but the final sample slightly deviates (e.g., 18.4% GPs). A brief discussion on whether this affected results would be helpful.

Geographic distribution (mostly France) may limit generalizability. Acknowledging this as a limitation is sufficient.

o Delphi Process:

The two-round Delphi approach with consensus meetings is rigorous.

The 75% agreement threshold is standard, but was there any consideration of weighting responses by expertise (e.g., diabetologists vs. pharmacists)?

4. Results

• Participant Demographics:

o Table 1 is clear, but the female percentage for GPs (14%) seems unusually low. Is this representative of the GP population in the region?

• Consensus Definitions:

o The 41 definitions are well-organized into four clinically relevant categories.

o Minor issue: Some definitions (e.g., #35 on metformin and dehydration) are highly specific. Were these derived from real-world cases or guidelines? A brief rationale for inclusion would help.

5. Discussion

• Strengths:

o Compares findings to existing PIP tools (e.g., Beers, STOPP/START).

o Highlights potential applications in clinical decision support systems (CDSS).

• Suggestions for Improvement:

o Clinical Relevance: Discuss how frequently these PIP scenarios occur in practice (e.g., are some rare?).

o Implementation Challenges:

How feasible are these definitions for automated CDSS use? Some may require complex data (e.g., HbA1c trends, patient-reported symptoms).

o Limitations:

Acknowledge that the list may not cover all ADs (e.g., thiazolidinediones, as noted).

The study focused on non-insulin ADs—could future work include insulin?

6. Conclusion

• Succinct and aligned with the study’s goals.

• Consider adding a sentence on next steps (e.g., validation in real-world settings).

6. PLOS authors have the option to publish the peer review history of their article (what does this mean? ). If published, this will include your full peer review and any attached files.

**Do you want your identity to be public for this peer review?** For information about this choice, including consent withdrawal, please see our Privacy Policy .

Reviewer #1: **Yes: ** Dr. Thelma Alalbila Aku

Reviewer #2: No

Reviewer #3: No

---

## [Author Response · Author response to Decision Letter 1]

6 Sep 2025

Response to Journal Requirement

Dear Editor,

We thank you for your careful review of our manuscript and for highlighting the additional requirements needed to comply with PLOS ONE standards. Please find below our detailed responses and the corresponding modifications made in the revised version.

1. PLOS ONE style requirements

We carefully reformatted the manuscript according to the PLOS ONE templates, including file naming conventions, structure, and referencing style.

Modifications:

Section headings (Level 1): increased from 16 pt bold → 18 pt bold

Section headings (Level 2): increased from 12 pt bold → 16 pt bold

Figure formatting: figure titles set in bold

2. Informed consent

We confirm that all participants provided electronic written informed consent before accessing the Delphi survey. They were informed that their responses would remain anonymous and had to actively confirm their agreement by checking a consent box. This procedure was reviewed and validated by the Data Protection Officer of University of Lille. We have clarified this in the Ethics approval of the Methods section.

Modifications:

Lines 109-112 in revised manuscript with track changes, sentence revised from:

“Participants were informed that their answers would remain anonymous and had to confirm that they did not object to the collection of their personal data”

To:

“All participants provided electronic written informed consent before participating in the Delphi survey. They were informed that their answers would remain anonymous and had to actively confirm their agreement by checking a consent box prior to accessing the questionnaire.”

3. Funding information in manuscript

We removed all funding information from the manuscript body, as requested.

4. Role of funders

We revised the Funding Statement to specify the role of funders. In our case, funders had no role in study design, data collection and analysis, decision to publish, or preparation of the manuscript. The amended Role of Funder statement is also included in the cover letter.

Modifications:

We added the following sentence in the cover letter:

5. Copyrighted Figure 1 (maps/satellite images)

We confirm that Figure 1 was created de novo in R (packages sf and tmap) using only openly licensed, with no Google or other proprietary basemaps. Country boundaries were obtained from Natural Earth (public domain). We have added a sentence in the Methods to document these sources and licenses. This complies with PLOS ONE’s CC BY 4.0 requirements.

Modifications:

Lines 183-184 in revised manuscript with track changes, we added the sentence follow:

“Fig. 1 was generated in R (sf, tmap) using Natural Earth administrative boundaries (public domain). No proprietary basemaps were used.”

6. Captions for Supporting Information

We added full captions for all Supporting Information files at the end of the manuscript and ensured that in-text citations match.

7. Reviewer-suggested citations

We carefully evaluated the suggested references. Relevant works have been cited in the revised manuscript; non-essential references were not added, in line with PLOS ONE’s policy.

8. Reference list

We thoroughly reviewed the reference list for completeness and accuracy. No retracted articles were cited.

Additional editor request: Grammar

We thank the editor for highlighting the importance of ensuring correct grammar. To address this, we have had the entire manuscript reviewed and edited by a professional scientific editor (David Fraser, Biotech Communication).  

Response to Reviewer 1 Comments

We thank Reviewer 1 for comments and constructive suggestions. We have carefully revised the manuscript according to each comment and suggestion. In the following, our responses are given in blue.

Comment: The paper was generally well written. Background: Captures the essentials of what the survey is about and the general aim of the paper. Methodology: Sound. Results and Discussion: Presented in a logical and simplified manner. Conclusion: It is a reflection of the aims and objectives that the authors sought to obtain. Authors should declare if they received funding or not for this work.

We sincerely thank the reviewer for the positive evaluation of our manuscript.

Regarding the funding declaration, we confirm that this research was supported by PreciDIAB, which is jointly funded by the French National Agency for Research (ANR-18-IBHU-0001), the European Union (FEDER – agreement NP0025517), the Hauts-de-France Regional Council (agreement 20001891/NP0025517), and the European Metropolis of Lille (MEL, agreement 2019_ESR_11). The funding statement has also been amended according to the Editors’ comments and PLOS ONE policy.

Modifications:

In accordance with PLOS ONE requirements, we have removed this information from the manuscript and ensured that it appears only in the Funding Statement section of the submission system. We have also included the following mandatory sentence on the role of funders in our cover letter:

Response to Reviewer 2 Comments

We thank Reviewer 2 for comments and constructive suggestions. We have carefully revised the manuscript according to each comment and suggestion. In the following, our responses are given in blue.

INTRODUCTION

Comment: Acknowledge that neither an implicit nor explicit approach to reducing potentially inappropriate medicines is a “fix all”.

We agree with the reviewer’s observation. It is obvious for all co-authors, but we agree that it should be written in an explicit way in the introduction. To address this, we have added a sentence in the Introduction to explicitly acknowledge that neither implicit nor explicit approaches can serve as a universal solution to inappropriate prescribing.

Modifications:

Lines 65-66 in revised manuscript with track changes, we added the following sentence in the revised manuscript with track changes:

“However, neither implicit nor explicit approaches should be considered as a universal solution to inappropriate prescribing.”

Comment: Include reference to personalization of treatment, which was inferred but not explicit.

We agree that personalization of treatment is an important concept and have revised the Introduction to explicitly highlight the importance of individualizing therapy according to patient characteristics.

Modifications:

Lines 66-67 in revised manuscript with track changes, we added:

“Implicit and explicit approaches should be combined, which underlines the importance of treatment personalization according to individual patient characteristics [1, 13].”

Comment (Line 57): “physicians and pharmacists face barriers to implementing guidelines...”- be explicit that you are referring to national and international guidance on prescribing of ADs.

We thank the reviewer for this suggestion. We have clarified the text to explicitly state that the barriers mentioned relate to the implementation of both national and international guidelines.

Modifications:

Line 56 in revised manuscript with track changes, sentence revised from:

“Physicians and pharmacists face barriers to implementing guidelines on the prescribing of antidiabetic drugs.”

to:

“Physicians and pharmacists face barriers to implementing national and international guidelines on the prescribing of antidiabetic drugs.”

METHODS

Comment (Lines 86–87): Please include that the explicit definitions were identified by literature as described in the study protocol.

We have clarified this point by specifying that explicit definitions were identified through a literature review and a qualitative study as outlined in the study protocol.

Modification:

Lines 96-98 in revised manuscript with track changes:

“The explicit definitions were identified through a systematic review of the literature and qualitative study, as described in the study protocol.”

Comment (Lines 106–107): How were these participants identified in the first instance? Were they from existing relationships within the steering group or did you reach out to special interest groups, etc.

Participants were identified through the professional networks of the steering committee. Each steering committee member reached healthcare professional who were specifically invested in AD prescribing and then contacted them by email.

Modifications:

Lines 118-121 in revised manuscript with track changes, sentence revised from:

“Participants were recruited via email.”

To:

“Participants were identified through the professional networks of the steering committee. Each steering committee member reached healthcare professional who were specifically invested in AD prescribing and then contacted them by email.”

Comment (Line 117): "the participants had to complete a questionnaire and indicate" refine to "participants completed a questionnaire and indicated.."

We have revisited the sentence as suggested.

Modifications:

Line 132 in revised manuscript with track changes, sentence revised from:

“The participants had to complete a questionnaire and indicate…”

To:

“Participants completed a questionnaire and indicated…”

Comment (Lines 122–132): Were participants provided with an individual report and the start of round 2 to inform them how their responses compared to others participating in the survey? Did you apply analyses e.g. a mixed effects regression model to the data. If not, why not?

We appreciate this question and have clarified our procedure. Participants did not receive individual feedback reports between rounds. Instead, for all items advanced to round 2, we presented aggregate feedback together with a synthesis of arguments during the first consensus meeting. This format enabled deeper discussion and more reflective re-assessment; colleagues who could not attend consensus meeting received the same quantitative summaries and a written synopsis of the discussion. To limit respondent burden, we deliberately did not embed these materials in the round-2 online questionnaire. We deliberately favored qualitative exchange and structured discussion over regression-based quantitative analyses, which, in our experience from previous Delphi studies, have limited impact on participants’ decisions. (https://pubmed.ncbi.nlm.nih.gov/38534718/,
https://pubmed.ncbi.nlm.nih.gov/29433501/)

Modification:

We add the following sentence in lines 135-139 in revised manuscript with track changes “Participants did not receive individual feedback reports between rounds. For items carried forward to round 2, we presented aggregate feedback together with a brief synthesis of arguments during the first consensus meeting to support reflective re-assessment. To minimise respondent burden, these materials were not embedded in the round-2 online questionnaire; non-attendees received the same quantitative summaries and a written synopsis of the discussion.”

Comment (Lines 155–156): "In other cases, the explicit definitions were considered to be indeterminate (neither validated156 nor rejected have been neither e in an state"... this sentence does not make sense.

We thank the reviewer for pointing out this ambiguity. The sentence has been rewritten to clearly explain that certain explicit definitions were considered indeterminate when neither validated nor rejected.

Modification:

Line 177 in revised manuscript with track changes, sentence revised from:

“In other cases, the explicit definitions were considered to be indeterminate (neither validated nor rejected have been neither in an state)”

To:

“In other cases, the explicit definitions were considered indeterminate (meaning that they were neither validated nor rejected by the panel).”

RESULTS

Comment (Table 1): Median year of qualification incorrectly reported. The table reports the median year, not the median years since qualification.

We agree with the reviewer and have corrected this error to accurately reflect that the table reports the median year of qualification, not the number of years since qualification.

Modifications:

In Table 1, we revised the label from:

“Median years since qualification [range]”

To:

“Median year of qualification [range]”

Comment (Figure 1): Figure 1. Whilst this is interesting to see which areas of France were represented in the Delphi study, it would also be interesting to see which healthcare professionals from these regions also participated. you report that you have one Swiss and one Belgian participant, but it would have been interesting to see what type of healthcare professional this was, e.g. GP, community pharmacist, etc.

We thank the reviewer for this suggestion. We have revised Figure 1 to include the distribution of participants by professional background (general practitioners, diabetologists, community pharmacists, hospital pharmacists, pharmacologist) and specified the professional category of the Swiss and Belgian participants.

Modification (Figure 1):

Figure 1. Geographic distribution of Delphi participants

Abbreviations: GP = general practitioner; Diab = diabetologist; C-Pharm = community pharmacist; H-Pharm = hospital pharmacist; Pharmacol = pharmacologist.

Comment (Line 175): I would open this paragraph describing figure 2 and then outline the timeframe. Currently the reference style made me think the figure also contained dates and not the flow of participants and the evolution of the definitions.

We have restructured the paragraph so that it first introduces Figure 2 before describing the timeframe of the Delphi process.

Modifications:

Lines 200 in revised manuscript with track changes, we added:

“Fig. 2 illustrates the flow of participants across the two Delphi rounds and the evolution of the explicit definitions over time.”

Comment (Line 183): Space required between the end of one sentence and start of another.

We corrected the typographical error by inserting the missing space.

Modifications:

Line 209 in revised manuscript with track changes revised from:

“2024.Of…”

To:

“2024. Of…”

DISCUSSION

Comment (Line 217): “Polypharmacy is very prevalent"- do not need “very".

We agree and have simplified the sentence by removing the unnecessary qualifier.

Modifications:

Line 275 in revised manuscript with track changes revised from:

“Polypharmacy is very prevalent among older adults with diabetes.”

To:

“Polypharmacy is prevalent among older adults with diabetes.”

Comment (Line 225): “Explicit tools for use with populations of older adults have been studied extensively"- give examples

We have added examples of widely used explicit tools, including the Beers criteria, STOPP/START, and the EU(7)-PIM list with references.

Modifications:

We modified Line 265 in revised manuscript with track changes as suggested:

“For example, the Beers criteria, STOPP/START, and the EU(7)-PIM list are widely used explicit tools for older adults [24, 25, 44-47].”

Comment (Lines 232–250): It feels more organic to lead with existing PIP tools then your tool, highlighting the differences.

We thank the reviewer for this constructive suggestion. We have reorganized the paragraph in the Discussion so that it now begins by describing existing explicit PIP tools (Beers, STOPP/START, EU(7)-PIM, PRISCUS), their benefits, and evidence of their impact. We then introduce our Delphi-based explicit list for antidiabetic drugs, situating our work as a complementary contribution in this established field. This reorganization improves the logical flow and makes the novelty of our contribution clearer.

Modifications:

Lines 264-283 in revised manuscript with track changes revised from:

“Our research study was prompted by studies in the field of geriatrics, in which PIPs were detected with explicit tools [24, 25, 40–43]. The explicit list obtained is specific for the management of ADs (excluding insulin) for people living with T2DM. Polypharmacy is very prevalent in people living with T2DM, due to the many comorbidities. Hence, the management of AD prescriptions in primary care can be challenging [5]. Faquetti et al.’s study of a prim

---

## [Decision Letter · Decision Letter 1]

2 Oct 2025

Development of explicit definitions of potentially inappropriate prescriptions for antidiabetic drugs in people with type 2 diabetes: a Delphi survey and consensus meeting.

PONE-D-25-27267R1

Dear Dr. Gerard,

We’re pleased to inform you that your manuscript has been judged scientifically suitable for publication and will be formally accepted for publication once it meets all outstanding technical requirements.

Kind regards,

Sairah Hafeez Kamran, PhD

Academic Editor

PLOS ONE

Additional Editor Comments (optional):

Reviewers' comments:

Reviewer's Responses to Questions

**Comments to the Author**

1. If the authors have adequately addressed your comments raised in a previous round of review and you feel that this manuscript is now acceptable for publication, you may indicate that here to bypass the “Comments to the Author” section, enter your conflict of interest statement in the “Confidential to Editor” section, and submit your "Accept" recommendation.

Reviewer #1: All comments have been addressed

Reviewer #2: All comments have been addressed

Reviewer #3: All comments have been addressed

2. Is the manuscript technically sound, and do the data support the conclusions?

Reviewer #1: Yes

Reviewer #2: Yes

Reviewer #3: Yes

3. Has the statistical analysis been performed appropriately and rigorously? 

Reviewer #1: Yes

Reviewer #2: Yes

Reviewer #3: Yes

4. Have the authors made all data underlying the findings in their manuscript fully available?

Reviewer #1: Yes

Reviewer #2: Yes

Reviewer #3: Yes

5. Is the manuscript presented in an intelligible fashion and written in standard English?

Reviewer #1: Yes

Reviewer #2: Yes

Reviewer #3: Yes

6. Review Comments to the Author

Reviewer #1: The authors have addressed all the outstanding issues.I identified typographical errors and they have addressed them.

Reviewer #2: Thank you for responding to and actioning comments from the reviewers. Your manuscript is more robust and better explains your processes and details of your research. However, there are a few minor typographical errors that have appeared in the manuscript when it was edited.

Line 127 "period" at the start of the sentence

Lines 309-311 -different font and size.

Reviewer #3: The authors have answered all my questions and concerns. I recommend that the manuscript be accepted.

7. PLOS authors have the option to publish the peer review history of their article (what does this mean? ). If published, this will include your full peer review and any attached files.

**Do you want your identity to be public for this peer review?** For information about this choice, including consent withdrawal, please see our Privacy Policy .

Reviewer #1: No

Reviewer #2: No

Reviewer #3: No

---

## [Editor Report · Acceptance letter]

PONE-D-25-27267R1

PLOS ONE

Dear Dr. Gerard,

I'm pleased to inform you that your manuscript has been deemed suitable for publication in PLOS ONE. Congratulations! Your manuscript is now being handed over to our production team.

Kind regards,

on behalf of

Dr. Sairah Hafeez Kamran

Academic Editor

PLOS ONE